# Association Between Blood Benzene Levels and Periodontal Disease in a Nationally Representative Adult U.S. Population

**DOI:** 10.3390/ijerph22060853

**Published:** 2025-05-29

**Authors:** Basel Hamoud, Meshari Alfailakwi, Hessah Aljalahmah, Fatema Almael, Sarah Alsaeedi, Khaled Saleh, Bushra Ahmad, Hend Alqaderi

**Affiliations:** 1Ministry of Health, Kuwait City 15462, Kuwait; baselmhamoud@gmail.com (B.H.); m.alfailakwi@hotmail.com (M.A.); dr.hessahaljalahma@hotmail.com (H.A.); fsalmael36x@gmail.com (F.A.); alsaeedisarah4477@gmail.com (S.A.); ksaleh.dds@gmail.com (K.S.); 2Department of Public Health, School of Dental Medicine, Tufts University, Boston, MA 02111, USA; bushra.ahmad@tufts.edu; 3Dasman Diabetes Institute, Kuwait City 15462, Kuwait

**Keywords:** periodontitis, oral health, benzene, cotinine, NHANES, biomarkers

## Abstract

(1) Background: Benzene, environmental pollutant, is linked to various adverse health effects, but its impact on oral health remains under-explored. This study examines the association between blood benzene levels and periodontitis, a progressive oral inflammatory condition, using a nationally representative sample of U.S. adults. (2) Methods: Cross sectional data from the 2013–2014 National Health and Nutrition Examination Survey (NHANES) were analyzed. Periodontitis was defined per CDC/AAP. Three weighted multivariable logistic regression models determined the association between blood benzene levels and periodontal severity, adjusting for potential confounders. A structural equation modeling (SEM) analysis evaluated cotinine, smoking biomarker, as a mediator in the relationship between benzene and severe periodontitis. (3) Results: The ordinal logistic regression showed a statistically significant association (AOR = 2.0, *p* = 0.02) between blood benzene levels and periodontal severity. A one unit increase in blood benzene was associated twice the odds of progressing to a higher category of periodontitis. Benzene exposure was significantly linked to severe periodontitis (AOR = 2.9, *p* = 0.001). SEM analysis indicated cotinine mediates the relationship between blood benzene and sever periodontitis. (4) Conclusions: This study provides evidence that higher blood benzene levels are associated with severe periodontitis. The findings suggest that cotinine, a biomarker of smoking, mediates the relationship between benzene exposure and severe periodontitis.

## 1. Introduction

Periodontal disease is a chronic inflammatory condition that affects the surrounding structures of the teeth, leading to progressive destruction of the periodontal ligament and alveolar bone [1]. It is a major public health problem, with severe periodontitis affecting approximately 10–15% of the global population [2]. In the United States, the prevalence of periodontitis among adults aged 30 years and older is estimated to be 42%, with 7.8% of the population having severe periodontitis [3]. Periodontal disease not only compromises oral health and function but also has been linked to various systemic conditions, such as diabetes, cardiovascular disease, and adverse pregnancy outcomes [4,5].

The etiology of periodontal disease is multifactorial, involving an intricate interplay between bacterial biofilms, host immune response, and environmental and genetic risk factors [6]. While the primary initiating factor is the accumulation of bacterial plaque, the host’s inflammatory response plays a critical role in the progression and severity of the disease [7]. In recent years, there has been growing interest in the potential role of environmental toxicants in the development and progression of periodontal disease, as these substances can modulate the host’s immune response and alter the composition of the oral microbiome [8].

Benzene, a volatile organic compound, is a ubiquitous environmental pollutant that has been identified as a significant public health concern [9]. It ranks among the top 20 chemicals produced by industrial sources in the United States, with annual emissions exceeding 6.7 million pounds [10]. It is commonly encountered through tobacco smoke, vehicle emissions, industrial solvents, and indoor chemical products [11]. In addition to industrial sources, mobile sources significantly contribute to atmospheric benzene levels, particularly near-point sources [11].

Low-dose benzene exposure is not limited to occupational or industrial settings. It is also encountered through common household products such as paints, air fresheners, and adhesives, particularly in poorly ventilated spaces [12]. Additional sources include gasoline vapors during refueling, wood stoves, incense, and indoor combustion [12]. Occupational exposure is prevalent in environments like nail salons, dry cleaners, and printing facilities. Human exposure is also significant via mainstream and secondhand cigarette smoke, with concentrations ranging from 35 to 70 ppm in mainstream smoke [13,14]. Due to its high volatility, inhalation is the primary route of human exposure to benzene [15]. Once inhaled or absorbed, benzene undergoes hepatic metabolism primarily via cytochrome P450 enzymes (notably CYP2E1), producing reactive intermediates such as benzene oxide, phenol, and hydroquinone [15]. These metabolites have been linked to oxidative stress, hematological toxicity, bone marrow impairment and increased risks of conditions such as aplastic anemia and acute myeloid leukemia [15]. Blood benzene is considered a reliable biomarker for evaluating recent exposure to benzene, particularly in occupational and environmental health contexts [16]. Its detection reflects exposure from multiple sources, including environmental, occupational, and behavioral factors [16]. Benzene has been shown to induce oxidative stress and trigger inflammation in various tissues [17,18]. Cotinine, the primary metabolite of nicotine, is frequently used to assess tobacco smoke exposure and correlates strongly with benzene levels in smokers [19].

While blood benzene testing is traditionally limited to occupational or hematologic evaluation, its potential relevance in oral health remains underexplored [20]. However, in this study, we evaluate benzene not just as a pollutant, but as a systemic toxicant with potential implications for periodontal tissue destruction. Using data from the National Health and Nutrition Examination Survey (NHANES), we investigate whether blood benzene levels are associated with periodontitis severity, thereby extending the clinical utility of benzene as a biomarker beyond its traditional applications in occupational or hematologic contexts.

Understanding blood benzene levels provides insight into cumulative exposure from sources such as tobacco smoke, household chemicals, and ambient air pollution. Given the well-established association between smoking and periodontal disease [21], and the correlation between blood benzene and cotinine levels [22,23], it was essential to account for smoking in this relationship. We utilized serum cotinine levels, an objective biomarker of both active and passive tobacco exposure, to explore whether cotinine mediates the association between benzene exposure and periodontitis. This study aims to contribute to a growing body of research exploring how environmental toxicants affect oral inflammatory conditions through systemic biological pathways.

## 2. Materials and Methods

### 2.1. Study Population

In this cross-sectional study, data was utilized from the 2013–2014 NHANES, a nationally representative survey of the non-institutionalized civilian population in the United States. The NHANES, conducted by the National Center for Health Statistics, Centers for Disease Control and Prevention, employs a complex probability sampling design involving multiple stages, stratification, and clustering. The NHANES sampling strategy includes stratification by demographic and geographic characteristics, cluster sampling within primary sampling units (PSUs), oversampling of specific population subgroups (e.g., non-Hispanic Black, Hispanic, Asian, and older adults) to ensure statistical reliability for these groups and use of sampling weights in analyses to adjust for unequal probabilities of selection, nonresponse, and post-stratification.

The NHANES collects data on various health outcomes and explanatory variables through a combination of interviews, laboratory tests, and clinical examinations. We used data from 2013–2014 survey cycle, approximately 5000 individuals of all ages were selected to participate. These participants were interviewed at their homes and then invited to undergo a comprehensive health assessment in Mobile Examination Centers. To minimize potential confounding factors related to temporal, examiner, and geographical variations, we restricted our sample to the most recently released cycle, 2013–2014. This approach is critical for our analysis of the environmental pollutants’ benzene and cotinine, ensuring consistency and reliability in our findings. As a component of the NHANES, trained and calibrated dental professionals conducted comprehensive periodontal examinations on survey participants aged 30 years and older within the Mobile Examination Centers. The study adhered to the Data Use Restrictions set forth by the National Center for Health Statistics. This study was conducted in accordance with the STROBE (Strengthening the Reporting of Observational Studies in Epidemiology) guidelines.

### 2.2. Eligibility Criteria

Inclusion Criteria:Adults aged ≥ 30 years, consistent with NHANES eligibility for full-mouth periodontal examinations.Participants with complete periodontal clinical data, including probing depth (PD) and clinical attachment loss (CAL).Participants with measured blood benzene and serum cotinine levels.Individuals with available data on key covariates: age, sex, race/ethnicity, education level, occupation, diabetes status.

Exclusion Criteria:Participants with missing data on periodontal status, blood benzene, or cotinine levels.Individuals with conditions requiring antibiotic prophylaxis prior to dental exams (excluded by NHANES protocols).Participants under 30 years old, since they were not eligible for the FMPE (Full-Mouth Periodontal Examination).Subjects with implausible biomarker values or incomplete demographic/medical history data.

### 2.3. Periodontal Examination

The NHANES study of 2013 to 2014 was conducted using a full-mouth periodontal examination (FMPE) among individuals aged ≥ 30 years who did not have a health condition that required antibiotic prophylaxis before periodontal test-ing. The FMPE was conducted with the intent to produce gold-standard assessments for clinical attachment loss (AL). For this reason, direct measurements of both the distance between the cemento-enamel junction and the free gingival margin (CEJ-FGM) and the probing depth (PD) were mea- sured at each site. All measurements were taken at six sites (mesiobuccal, midbuccal, distobuccal, mesiolingual, midlin-gual, and distolingual) of all teeth with the exclusion of third molars. All calculations were rounded to the lower whole millimeter. Clinical AL was calculated based on these two measurements.

### 2.4. Definition of the Dependent Variable: Severe Periodontitis

Periodontitis was used as categorical variable with four categories: no periodontitis, mild periodontitis, moderate periodontitis, and severe periodontitis. The categorization was based on clinical attachment loss (CAL) and probing pocket depth (PPD) in accordance with the latest guidelines from the American Academy of Periodontology (AAP) and the Centers for Disease Control and Prevention (CDC) [22]. Severe periodontitis was defined as having two or more interproximal sites with CAL of 6 mm or greater and one or more interproximal sites with PPD of 5 mm or greater. Moderate periodontitis was characterized by having two or more interproximal sites with CAL of 4 mm or greater or two or more interproximal sites with PPD of 5 mm or greater. Mild periodontitis was defined as having two or more interproximal sites with CAL of 3 mm or greater and two or more interproximal sites with PPD of 4 mm or greater (not on the same tooth) or one or more sites with PPD of 5 mm or greater.

### 2.5. Description of Independent Variable: Benzene Exposure

In this study, benzene exposure is quantified as a continuous variable. Blood benzene concentrations were measured in whole blood using gas chromatography coupled with high-resolution mass spectrometry (GC-HRMS), with a lower limit of detection of 0.06 ng/mL [24]. The method specifically targets volatile organic compounds (VOCs) in whole blood. Blood was drawn into a gas-tight glass vacutainer immediately refrigerated and then frozen at −70 °C until analysis to prevent volatile loss. NHANES included calibration standards, field blanks, and QC samples in every batch. Based on our NHANES 2013–2014 data, we analyzed blood benzene (ng/mL), not SPMA (S-phenylmercapturic acid), which is a urinary metabolite. In this study, benzene exposure was assessed using blood benzene concentration, reported in nanograms per milliliter (ng/mL), as measured in whole blood by NHANES. There was no data on SPMA levels in NHANSE in this cycle.

Samples were analyzed at the CDC National Center for Environmental Health (NCEH) VOC Laboratory [25].

### 2.6. Cotinine: Mediator Variable

We investigated the role of cotinine as a potential mediator in the relationship between benzene exposure and severe periodontitis. Cotinine, a metabolite of nicotine, is a reliable biomarker of tobacco smoke exposure. Our study included both smokers and non-smokers. Smoking status was not used as an inclusion criterion. Instead, serum cotinine levels were used as a continuous biomarker to objectively capture exposure to both active and passive tobacco smoke. Blood samples were collected from participants and centrifuged, serum isolated, and stored at −20 °C or lower until analysis. Serum cotinine levels were measured using high-performance liquid chromatography/tandem mass spectrometry (LC/MS/MS), a highly sensitive analytical technique that separates, identifies, and quantifies compounds by combining liquid chromatography (LC) for separation and tandem mass spectrometry (MS/MS) for detection and analysis. In the NHANES 2013–2014 cycle, serum cotinine levels were measured using isotope-dilution high-performance liquid chromatography with atmospheric pressure chemical ionization tandem mass spectrometry (ID HPLC-APCI MS/MS) with a detection limit of 0.015 ng/mL. The instrument used was the Shimadzu Nexera HPLC and AB Sciex API 6500 mass spectrometer. Only serum cotinine values were available in NHANES 2013–2014 cycle. Serum cotinine is considered the gold-standard biomarker for tobacco exposure due to its higher accuracy and stability.

Both benzene and cotinine biomarkers were analyzed at the CDC’s Division of Laboratory Sciences using standardized NHANES protocols with rigorous quality control [23]. NHANES (National Health and Nutrition Examination Survey) is a program conducted by the National Center for Health Statistics (NCHS) that collects data prospectively through standardized protocols during its cross-sectional survey cycle. We analyzed already collected, publicly available data of 2013–2014 NHANES dataset [23].

### 2.7. The Potential Confounding Variables

In this study, several demographic and health-related variables were categorized for analysis. Age was grouped into four categories: 30–34 years, 35–49 years, 50–64 years, and 65 years or older. Sex was classified as a binary variable (male or female) while race was treated as a nominal variable with categories including “Non-Hispanic White”, “Non-Hispanic Black”, “Hispanic”, “Non-Hispanic Asian”, and “Other” races. Education level was divided into three categories: less than high school, high school graduate or equivalent (including GED), and more than high school education. Occupation was determined based on current employment status and job type, with individuals classified as either occupied (currently employed or self-employed) or not occupied (unemployed or not seeking employment). Diabetes status was assessed using a single question, “Have you ever been told by a doctor or health professional that you have diabetes or sugar diabetes?” and responses were categorized into two groups: “no” for no diabetes and “yes” or “borderline” for the presence of diabetes or pre-diabetes.

### 2.8. Statistical Methods

We conducted Chi-square tests to compare categorical demographic variables stratified by the disease categories: no periodontitis, mild, moderate, and severe periodontitis. Additionally, we used a *t*-test to assess the difference in benzene and cotinine levels between subjects in these categories.

To investigate the association between benzene exposure and periodontitis, four weighted multivariate logistic regression models were conducted. An ordinal logistic regression model was performed with periodontal status as the dependent variable, which has four categories: no periodontitis, mild periodontitis, moderate periodontitis, and severe periodontitis, defined according to specific clinical criteria related to attachment loss and probing depth, as previously described. Independent variables included the exposure variable, benzene, while confounding variables comprised age, sex, race, education level, occupation, medical history, and diabetes. Additionally, three separate weighted multiple logistic regression analyses were conducted for the three binary periodontitis outcomes: mild, moderate, and severe periodontitis, adjusting for the aforementioned confounders.

Adjusted odds ratios (AORs) were reported with their corresponding 95% confidence intervals (CIs) and *p*-values, considering a 0.05 significance level. NHANES survey weights were incorporated into the analyses to account for the complex sampling design and to provide estimates representative of the U.S. population.

### 2.9. Structural Equation Modeling (SEM) Mediation Analysis

To explore the mediating role of smoking (measure by blood cotinine level) in the relationship between benzene exposure and severe periodontitis, we conducted a Structural Equation Modeling (SEM) mediation analysis. SEM is an advanced statistical technique that enables the examination of complex relationships among multiple variables, incorporating both direct and indirect effects. We aimed to assess the potential mediating effect of smoking, represented by cotinine levels, a subjective unbiased measure of both first and secondhand smoking, in this causal pathway. The SEM model was specified to include the direct effect of benzene exposure on severe periodontitis, the direct effect of benzene exposure on cotinine levels, and the direct effect of cotinine levels on severe periodontitis. We also examined the indirect effect of benzene exposure on severe periodontitis mediated by cotinine levels.

In our study, we used log10 transformation for blood benzene and cotinine measurements to account for the skewed distribution of the data. All statistical analyses were performed using STATA 17 software (StataCorp LLC, College Station, TX, USA).

## 3. Results

Table 1 presents a descriptive summary of the study population’s characteristics, stratified by the categories of periodontal status.

Severe periodontitis was more prevalent in older adults (≥50 years), males, Non-Hispanic Black and Hispanic populations and Individuals with lower education and income levels. Gender distribution shows a higher proportion of males in severe periodontal disease cases (10.33%) compared to females (4.08%), with a *p*-value < 0.001 indicating statistical significance. Age groups also show significant variation, with the highest prevalence of severe periodontal disease in individuals aged 45–64 (11.49%) and those aged 65+ (6.17%), again with a *p*-value < 0.001. Racial differences are evident, with 11.18% of Black participants and 8.83% of Hispanic participants having severe periodontal disease. Education level is inversely related to the prevalence of severe periodontal disease, with those having less than high school education showing the highest prevalence (11.80%).

Table 2 shows that mean benzene and cotinine levels are higher in individuals with severe periodontitis. Severe periodontitis is associated with a mean benzene level of 0.11 ng/mL (SD = 0.15) versus 0.06 ng/mL (SD = 0.11) in those without (*p* = 0.001). Cotinine levels are also higher in severe periodontitis cases (mean = 124.86 ng/mL, SD = 173.54) compared to those without (mean = 57.35 ng/mL, SD = 135.22) with a *p*-value of 0.001. Similar trends were observed for mild and moderate periodontitis groups, indicating a strong correlation between higher benzene and cotinine levels and increased periodontal disease severity.

The standard deviations for blood benzene and serum cotinine levels exceeded their respective means, suggesting a right-skewed distribution. This skewness indicates that a small subset of individuals may have significantly elevated levels compared to the general population. To address this, a log10 transformation was applied to both benzene and cotinine levels prior to analysis.

Due to NHANES subsampling protocols, it doesn’t test every biomarker for every participant. Instead, blood benzene levels were measured in a one-third random subsample of eligible participants, resulting in a smaller sample size (n = 2096) compared to serum cotinine levels, which were assessed in a broader segment of the population (n = 4464). Furthermore, because Benzene testing is expensive and requires specialized procedures (due to its volatility), it’s often limited to smaller, more tightly controlled subsamples [26]. The difference in sample sizes reflects standard NHANES lab testing design and does not indicate exclusion due to study-specific criteria.

Table 3 presents the four multiple logistic regression model examining the association between benzene exposure and periodontal disease severity. The adjusted odds ratio (AOR) of 2.0 (95% CI: 1.12–3.9, *p* = 0.02) from the ordinal logistic regression indicates a statistically significant association between blood benzene levels (measured in ng/mL) and the severity of periodontal disease. For every one-unit increase in blood benzene levels, the odds of being in a higher category of periodontal disease (i.e., moving from no periodontitis to mild, from mild to moderate, or from moderate to severe periodontitis) increase by 2 times.

Findings from the logistic regression models indicate that benzene exposure is significantly associated with severe periodontitis, with an AOR of 2.9 (95% CI: 1.6–5.3, *p* = 0.001), suggesting that a one-unit increase in log-transformed benzene levels is associated with a nearly three-fold increased likelihood of having severe periodontal disease, after controlling for other relevant factors. However, benzene is not significantly associated with mild (AOR: 1.732, 95% CI: 0.9–3.06, *p* = 0.057) or moderate periodontitis (AOR: 1.248, 95% CI: 0.6–2.1, *p* = 0.465). Age was found to be a significant predictor of severe periodontal disease, with older age groups exhibiting progressively higher adjusted odds ratios compared to the reference group (age 30 to 34). Non-Hispanic Black individuals had a significantly higher likelihood of severe periodontal disease compared to non-Hispanic White Americans (AOR = 3.65, *p* = 0.002), while education level showed a protective effect, with individuals having more than a high school education displaying significantly lower odds of severe periodontal disease compared to those with less than a high school education (AOR = 0.46, *p* = 0.044).

Although diabetes is not statistically significant in the model, it is retained because it is a known confounder variable. Including it helps control for potential bias and provides more accurate estimates of the effects of other predictors. Diabetes is also likely correlated with one or more other variables in the model, which may suppress its apparent statistical significance.

Table 4 presents the results of a Structural Equation Modeling (SEM) analysis conducted to explore the mediation effects of cotinine levels in the relationship between benzene exposure and severe periodontitis. SEM analysis demonstrated that benzene levels have a statistically significant positive association with cotinine levels (estimate = 2.93, SE = 0.05, *p* < 0.001, 95% CI: 2.83 to 3.02), the direct relationship between benzene levels and severe periodontitis is not statistically significant (estimate = 0.29, SE = 0.25, *p* = 0.261, 95% CI: −0.21–0.78), and cotinine levels show a statistically positive association with severe periodontitis (estimate = 0.23, SE = 0.07, *p* = 0.001, 95% CI: 0.09–0.37). The Structural Equation Modeling (SEM) analysis suggests that benzene exposure increases cotinine levels, and higher cotinine levels, in turn, increase the risk of severe periodontitis. However, the direct relationship between benzene exposure and severe periodontitis is not statistically significant, indicating that the effect of benzene on severe periodontitis is likely mediated through its impact on cotinine levels. Directed Acyclic Graph (DAG) illustrates the mediation effect of cotinine in the relationship between benzene and severe periodontal disease (Figure 1).

## 4. Discussion

The present study investigated the association between blood benzene levels and periodontitis in a large, nationally representative sample of U.S adults. Our findings suggest that higher blood benzene levels are more likely to have severe periodontitis, a progressive inflammatory oral condition. Benzene, a ubiquitous environmental pollutant, has been linked to various adverse health outcomes, including hematological disorders, cancer, and immune system dysfunction [27]. However, its potential impact on periodontal health has not been investigated. Our study provided new evidence supporting the association between benzene exposure and periodontitis.

There’s minimal engagement with prior studies linking environmental toxicants to periodontitis, beyond generic mentions of toxicants or oxidative stress. Also, there’s no specific comparison to studies that might’ve explored benzene exposure in relation to periodontal outcomes. Most prior research has focused on air pollutants such as particulate matter (PM2.5), ozone (O_3_), and nitrogen dioxide (NO_2_), which have been associated with elevated risks of periodontal disease and systemic inflammation [28]. Studies reported that exposure to PM2.5 was significantly associated with increased odds of periodontitis [29,30]. Similarly, other studies observed an association between residential air pollution and attachment loss, a clinical marker of periodontal destruction [29,30]. These studies support that environmental exposures can induce oxidative stress, pro-inflammatory cytokine production, and host immune modulation which are all critical pathways in periodontal pathogenesis [28,29,30,31].

This study fills that gap by using blood benzene as an objective exposure marker and shows that cotinine may mediate the link between benzene and periodontitis.

In the human body, cotinine and benzene are closely interrelated due to their common presence in tobacco smoke, which serves as a major source of exposure to both compounds [32]. Cotinine, the primary metabolite of nicotine, is widely used as a biomarker of tobacco exposure, including both active and passive smoking [32]. Similarly, benzene, a volatile organic compound and known carcinogen, is released in substantial quantities during the combustion of tobacco products [33]. Studies have shown that mainstream cigarette smoke contains 35–70 parts per million (ppm) of benzene, and exposure levels are significantly elevated in smokers compared to non-smokers [33].

The metabolic pathways of nicotine and benzene are distinct, but their systemic presence often overlaps in smokers [34,35]. While nicotine is metabolized primarily in the liver by CYP2A6 into cotinine, benzene is metabolized by CYP2E1 into reactive intermediates that can generate oxidative stress and inflammation [34,35]. Additionally, in population-based biomonitoring studies, individuals with higher cotinine levels, typically smokers, also exhibit elevated benzene metabolite levels, indicating that smoking is a common source of exposure to both compounds [36,37]. Therefore, this correlation provides a mechanistic and epidemiological justification for using cotinine as a mediator in modeling the relationship between benzene and periodontitis, as we have done in the current study.

Studies have shown that metabolic activation of benzene in the human body is primarily mediated by cytochrome P450 enzymes, particularly CYP2E1, which converts benzene into reactive intermediates such as benzene oxide and benzoquinones [38]. These metabolites are known to induce oxidative stress, damage hematopoietic and epithelial tissues, and stimulate the production of proinflammatory cytokines like TNF-α and IL-1β [38]. In scenarios of chronic low-dose exposure or genetic polymorphisms affecting CYP450 function, these enzymes may become overwhelmed or dysregulated, leading to heightened systemic inflammation and immune alteration [38]. This pathway may help explain the observed association between blood benzene levels and periodontitis in our study. Similarly, the metabolism of nicotine into cotinine by CYP2A6 varies by individual, and may influence both exposure levels and inflammatory outcomes, further justifying cotinine role as a mediator [39].

Although benzene is a volatile compound primarily absorbed via inhalation, it is rapidly distributed through the bloodstream due to its high lipid solubility, enabling it to reach multiple tissues, including the oral cavity [40]. While direct measurement of benzene in gingival crevicular fluid or periodontal pockets is limited, its presence in blood suggests it can diffuse into inflamed or vascularized periodontal tissues, potentially contributing to local oxidative stress and inflammatory signaling [41]. The periodontal pocket itself, characterized by ulcerated epithelium and increased vascular permeability, may act as a microenvironmental reservoir, allowing systemically circulating toxicants like benzene and cotinine to accumulate or exert localized effects [41,42,43].

Furthermore, there may be a biochemical interaction or additive effect between benzene and cotinine in stimulating inflammatory pathways. Both compounds are metabolized via cytochrome P450 enzymes and are known to promote oxidative stress, proinflammatory cytokine release, and immune modulation [44]. Cotinine has been shown to enhance production of TNF-α, IL-1β, and IL-6 in periodontal tissues, while benzene metabolites like hydroquinone are linked to DNA damage and immune dysregulation [42,44,45]. Studies have shown that increased cotinine levels, as an objective biomarkers of tobacco exposure, contribute to periodontal tissue destruction by generating oxidative stress, impairing immune cell function, and disrupting fibroblast activity, which promote the recruitment of immune cells to the site of infection and all of which promotes loss of periodontal attachment [41,46]. Studies show that serum or salivary cotinine is more reliable than self-reported smoking status in predicting the severity and risk of periodontitis as it is present in serum and persists for a longer time compared to nicotine, with a half-life of about 19 h [47]. Although no direct synergistic mechanism between benzene and cotinine has been fully elucidated, their shared presence in tobacco smoke and overlapping biological effects suggest the possibility of amplified inflammatory responses when both are elevated, an area worthy of further exploration in periodontal research.

We utilized blood cotinine as an objective marker of both first and second-hand smoking, to avoid recall bias.

Oxidative stress can lead to the activation of redox-sensitive transcription factors, such as nuclear factor-kappa B (NF-κB), which regulate the expression of pro-inflammatory cytokines and matrix metalloproteinases involved in periodontal tissue destruction [48]. Additionally, benzene exposure has been associated with alterations in immune function, including decreased lymphocyte counts and impaired T-cell function, which may compromise the host’s ability to mount an effective defense against periodontal pathogens [49]. The relationship between benzene exposure and periodontal disease may be mediated by several mechanisms [50]. Given the strong link between benzene exposure, smoking, and periodontal disease [41,51], it was critical to address the role of smoking in the relationship between benzene and periodontal disease. The CDC’s Fourth National Report on Human Exposure to Environmental Chemicals, derived from NHANES laboratory data, reports a median blood benzene concentration of approximately 0.09 ng/mL among non-smokers, with a 95th percentile value of 0.49 ng/mL. These figures provide a benchmark for evaluating environmental benzene exposure in the absence of tobacco use [52].

In NHANES-based analyses reported by the CDC, smokers exhibit significantly elevated blood benzene concentrations, with median levels around 0.42–0.60 ng/mL and 95th percentile values exceeding 1.00 ng/mL. These levels are 5 to 10 times higher than those observed in non-smokers, confirming that tobacco smoke is a major source of benzene exposure in the general population [52].

Cigarette smoke contains benzene, and exposure to this toxin through smoking can lead to increased oxidative stress and inflammation in the oral cavity [41,53]. Oxidative stress has been implicated in the destruction of periodontal tissues, while inflammation can promote the progression of periodontal disease [54]. Furthermore, smoking has been associated with alterations in the oral microbiome, promoting the growth of periodontal pathogens such as Porphyromonas gingivalis and Tannerella forsythia [55,56]. These changes in the oral microbiome, coupled with the suppression of the host immune response due to smoking, can create a favorable environment for the development and progression of periodontal disease [56]. The impaired immune function may hinder the body’s ability to combat the pathogenic bacteria and maintain periodontal health [57].

The combination of these bacterial pathogens with the inflammatory and immune-suppressive effects of benzene exposure creates a vicious cycle that leads to the breakdown of periodontal tissues and the advancement of periodontitis [55,56,57,58]. Our study demonstrated that cotinine levels mediate the association between benzene exposure and periodontitis. In other words, there was an indirect effect of smoking on the relationship between benzene exposure and severe periodontitis. This suggests that smoking may play a critical role in the pathway leading to periodontal disease among individuals exposed to benzene.

### 4.1. Limitation of the Research

The strengths of our study include the use of a large, nationally representative sample, the objective assessment of periodontal status using standardized clinical measures, and the measurement of blood benzene levels as a biomarker of exposure. However, some limitations should be acknowledged. Although a significant association between benzene exposure and severe periodontitis was identified, the cross-sectional nature of the data does not allow us to definitively conclude whether higher benzene levels are a direct cause of periodontal disease or whether both may be influenced by other confounding factors. Additionally, the use of blood benzene levels as a biomarker of exposure may not fully capture long-term or cumulative benzene exposure. Blood benzene levels reflect recent exposure, but a more comprehensive understanding of the long-term impact on periodontal health may require measures of cumulative exposure over time. As such, our findings may not fully represent the relationship between chronic benzene exposure and the development of periodontitis. Another limitation is the potential for residual confounding. It is important to recognize that variations in serum benzene and cotinine levels may arise from a variety of sources beyond smoking status. Factors such as occupational exposure, ambient air pollution, indoor chemical use, and genetic polymorphisms involved in detoxification pathways can influence benzene metabolism and distribution. Similarly, cotinine levels are shaped not only by smoking behavior but also by passive smoke exposure, use of nicotine replacement therapies, metabolic rate, and individual genetic differences in nicotine metabolism. These variables may introduce heterogeneity into biomarker measurements and could serve as potential confounders or effect modifiers in the observed associations. While we adjusted for several key demographic and behavioral factors, there may still be unmeasured variables influencing the relationship between benzene and periodontitis.

Additionally, smoking behavior was indirectly assessed using cotinine levels, and any misclassification or underreporting of smoking status could have introduced bias into our result. Although NHANES collects detailed self-reported information on tobacco use habits, including quantity and duration of smoking, type of tobacco product used (e.g., cigarettes, cigars, hookah), and location of exposure, our study used serum cotinine levels as an objective biomarker of tobacco exposure. This approach minimizes recall bias and captures both active and passive exposure, but it limits our ability to differentiate between specific smoking behaviors, frequency, or modes of intake. Future studies may benefit from integrating both biomarker data and self-reported smoking behaviors to provide a more understanding of tobacco-related exposure patterns.

### 4.2. Future Perspectives

Future research should aim to address the limitations of this study and further explore the relationship between benzene exposure and periodontitis. Longitudinal cohort studies would be invaluable in establishing a temporal relationship between benzene exposure and the progression of periodontal disease. Such studies would track participants over time, allowing researchers to observe how changes in benzene exposure influence the onset and progression of periodontitis, rather than merely providing a snapshot of associations at a single point in time.

In a cohort study, individuals could be recruited based on their baseline exposure to benzene, whether occupational, environmental, or through smoking, and then followed over several years.

Additionally, future studies could explore the effects of other environmental pollutants, particularly in combination with benzene exposure. Examining the synergistic effects of multiple pollutants could provide a more comprehensive understanding of how environmental factors collectively contribute to the onset and progression of periodontitis. This approach would allow researchers to assess the cumulative burden of pollution on oral health, beyond the impact of benzene alone. Moreover, further investigation into the biological mechanisms underlying the observed association is essential. Future studies could delve into the role of oxidative stress, inflammatory pathways, and immune system modulation in mediating the effects of benzene on periodontal tissues. Identifying specific biomarkers associated with benzene exposure could also aid in early detection and intervention strategies for individuals at higher risk of periodontitis.

## 5. Conclusions

Our study provides new evidence that elevated blood benzene levels are associated with an increased risk of severe periodontitis in U.S. adults. Notably, we found that cotinine, a biomarker of smoking, mediates the relationship between benzene exposure and severe periodontitis. These findings emphasize the need to account for environmental factors, such as benzene exposure, in both the prevention and management of periodontal disease. For clinicians, this means recognizing environmental and occupational exposures as potential contributors to oral inflammation, especially in vulnerable groups like smokers, urban populations, and industrial workers. Benzene, often overlooked in dental practice, could be a hidden driver of advanced periodontal breakdown. For policymakers and public health officials, our findings support the integration of environmental health surveillance into oral health risk assessments. This includes strengthening tobacco control, air quality regulations, and workplace safety standards to reduce benzene exposure. Incorporating biomarkers like cotinine into dental screenings may help identify at-risk individuals who could benefit from early interventions and counseling on environmental risks. Public health strategies that reduce pollutant exposure and promote smoking cessation could have a significant impact on periodontal health. Future longitudinal studies are needed to explore the biological mechanisms involved and to examine potential synergistic effects of multiple environmental pollutants on periodontal disease.

## Figures and Tables

**Figure 1 ijerph-22-00853-f001:**
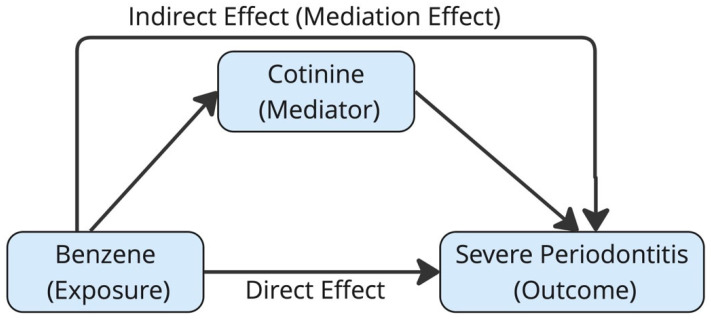
Directed Acyclic Graph (DAG) illustrating the mediation effect of cotinine in the relationship between benzene and severe periodontal disease.

**Table 1 ijerph-22-00853-t001:** Descriptive summary of population characteristics: Comparison with and without severe periodontal disease.

Covariate	No Periodontitis N = 2930	Mild Periodontitis N = 614	Moderate Periodontitis N = 796	Severe Periodontitis N = 329	Total N = 4669	*p*-Value
Sex						<0.001
male	1221 (55.10%)	346 (15.61%)	420 (18.95%)	229 (10.33%)	2216 (100%)	
female	1709 (69.67%)	268 (10.93%)	376 (15.33%)	100 (4.08%)	2453 (100%)	
Age						<0.001
30–34	354 (70.66%)	93 (18.56%)	41 (8.18%)	13 (2.59%)	501 (100%)	
35–49	982 (66.94%)	247 (16.84%)	165 (11.25%)	73 (4.98%)	1467 (100%)	
50–64	817 (56.89%)	168 (11.70%)	286 (19.92%)	165 (11.49%)	1436 (100%)	
65+	777 (61.42%)	106 (8.38%)	304 (24.03%)	78 (6.17%)	1265 (100%)	
Race	1402 (68.73%)	190 (9.31%)	351 (17.21%)	97 (4.75%)	2040 (100%)	<0.001
white	539 (56.86%)	150 (15.82%)	153 (16.14%)	106 (11.18%)	948 (100%)	
Black Hispanic	574 (56.33%)	194 (19.04%)	161 (15.80%)	90 (8.83%)	1019 (100%)	
Asian	341 (62.34%)	69 (12.61%)	107 (19.56%)	30 (5.48%)	547 (100%)	
Other	74 (64.35%)	11 (9.57%)	24 (20.87%)	6 (5.22%)	115 (100%)	
Education						<0.001
<High School	563 (54.03%)	174 (16.70%)	182 (17.47%)	123 (11.80%)	1042 (100%)	
High School/GED	593 (56.64%)	157 (15.00%)	202 (19.29%)	95 (9.07%)	1047 (100%)	
College or More	1772 (68.79%)	282 (10.95%)	411 (15.95%)	111 (4.31%)	2576 (100%)	
Diabetes						0.927
No	2524 (63.66%)	521 (13.14%)	641 (16.17%)	279 (7.04%)	3965 (100%)	
Yes	403 (57.49%)	93 (13.27%)	155 (22.11%)	50 (7.13%)	701 (100%)	
Any disease						0.072
No	1626 (63.32%)	389 (15.15%)	357 (13.90%)	196 (7.63%)	2568 (100%)	
Yes	1294 (62.00%)	224 (10.73%)	438 (20.99%)	131 (6.28%)	2087 (100%)	

**Table 2 ijerph-22-00853-t002:** Comparison of mean and standard deviation (SD) of benzene and cotinine levels across four periodontal categories. (ANOVA was used to assess significant differences between the means across the groups.

Measurement Periodontal Status Benzene (ng/mL)	Benzene (ng/mL) (N = 2096)			Cotinine (ng/mL) (N = 4464)		
	Mean (SD)	Sample Number	*p*-Value	Mean (SD)	Sample Number	*p*-Value
No periodontitis	0.05 (0.1)	1288	<0.0001	48.8 (126.6)	2790	<0.0001
Mild periodontitis	0.07 (0.1)	293	<0.0001	76.7 (144.3)	588	<0.0001
Moderate periodontitis	0.07 (0.1)	364	<0.0001	73.4 (153.9)	771	<0.0001
Severe periodontitis	0.1 (0.1)	151	<0.0001	124.8 (173.5)	315	<0.0001
Total	2096			4464		

**Table 3 ijerph-22-00853-t003:** Multiple logistic regression model for the association between severe periodontitis and benzene exposure.

		Composite		
Covariate	Odds Ratio	Confidence Interval		*p* Value
		Lower	Upper	
Benzene	2.9	1.6	5.3	(0.001) *
Age				
35 to 49	6.8	1.1	40.8	(0.037) *
50 to 64	15.5	2.8	86.4	(0.004) *
≥65		2.6	298.8	(0.009) *
Sex (reference: male)	0.5	0.2	1.2	(0.146)
Race (reference: non-Hispanic white)				
Non-Hispanic black	3.2	1.4	6.8	(0.005) *
Hispanic	1.5	0.3	7.4	(0.545)
Non Hispanic Asian	1.8	0.5	6.1	(0.285)
Other	6.0	0.7	49	(0.085)
Education (reference: <high school)				
High school/GED	0.9	0.2	2.8	(0.084)
Some college or more	0.4	0.2	1.0	(0.060)
Occupation	1.2	0.5	2.7	(0.612)
Diabetes	1.1	0.3	4.1	(0.824)
Any disease	0.2	0.09	0.8	(0.025) *

* *p* value < 0.05.

**Table 4 ijerph-22-00853-t004:** Structural Equation Modeling (SEM) results for the mediation analysis of the relationship between benzene exposure, cotinine levels, and severe periodontitis.

Path	Estimate	Standard Error	*p*-Value	95% CI
Benzene > Cotinine	2.93	0.05	< 0.001	2.83–3.02
Benzene > Severe periodontitis	0.29	0.25	0.261	−0.21–0.78
Cotinine > Severe periodontitis	0.23	0.07	0.001	0.09–0.37

## Data Availability

The data used in this article are publicly available and can be found at the Centers for Disease Control and Prevention (CDC) National Center for Health Statistics: National Health and Nutrition Examination Survey (NHANES) Questionnaires, Datasets, and Related Documentation, available at the following website: Accessed 4 December 2023 (https://wwwn.cdc.gov/nchs/nhanes/) [59].

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
