# Peer review of "Association Between Blood Benzene Levels and Periodontal Disease in a Nationally Representative Adult U.S. Population"

_ijerph, 2025, doi:10.3390/ijerph22060853_

Round 1

Reviewer 1 Report

Comments and Suggestions for Authors

The subject of the paper is of interest and relevant. A sufficiently large sample size has been evaluated.

The following queries may be looked into for clarification and added value.

  1. Describe the interrelation between cotinine and benzene in the human body.
  2. What were the inclusion and exclusion criteria for patients.
  3. What are the other sources of low dose chronic exposure to benzene?
  4. What was the sampling strategy used.
  5. Were STROBE guidelines followed.
  6. Were only smokers included, as line 176 mentions second hand smoking.
  7. What information was collected regarding the habit? of cigarettes smoked per day, duration, any filters used, type of smoke inhalation – cigars, pipes, hookah, vapes; in a private smoking zone [in office or hookah bars], personal space or public places, other forms of tobacco intake
  8. What were the methods for sampling benzene and cotinine?
  9. Were salivary cotinine values considered also.
  10. Why is there a difference in the number of patients sampled for benzene and cotinine (table 2)?
  11. How much of benzene is present in the blood of non-smokers never exposed to tobacco? Quote evidence reference range for comparison?
  12. Are any cut offs suggested in this study to indicate passive / active smokers ?
  13. Were retrospective data retrieved for serum benzene and serum cotinine (line 131).
  14. “The SPMA concentration is reported in either micrograms per liter (μg/L) or normalized 122 to creatinine levels (μg/g creatinine), serving as a biomarker for recent benzene exposure” – Line 122-123. Was the data for benzene reported as SPMA levels or creatinine levels. Please explain.
  15. Table 2 – Please account for higher S.D than mean for both benzene and cotinine.
  16. Line 357 –“Such studies would Longitudinal stud-“….Please correct.
  17. Line 397 – states human subjects were not involved – Then how was the data on benzene and cotinine in periodontitis patients obtained to suggest similarity of cohort in terms of methodology and reasons for blood sampling of benzene or SPMA.
  18. What other factors can cause variations in serum benzene and cotinine levels.
  19. Literature review and discussion on similar research work is lacking.
  20. What is the message the authors are trying to convey. How will it help clinicians or policy makers.

Author Response

Thank you for your valuable feedback and insightful comments. Your input has helped us improve the clarity and quality of the manuscript.

We have carefully revised the manuscript in response to each reviewer's suggestions. All changes are highlighted in yellow in the revised manuscript and in blue in the responses below. We provide detailed replies to every comment for your review in a pdf document.

Sincerely,

Basel Hamoud

Reviewer 2 Report

Comments and Suggestions for Authors

I greatly appreciate the opportunity to review this intriguing study assessing benzene as a potential contributor to the progression of periodontal disease. I would like to offer a few comments that may help improve the quality of the manuscript.

1.- The authors clearly delineate the shared signaling pathways among benzene, cotinine, and periodontal disease in amplifying the inflammatory response contributing to disease progression. Nevertheless, I would encourage the authors to address the role of cytochrome P450, as its function would likely need to be circumvented or compromised to allow for the upregulation of proinflammatory cytokines.

2.- Clarifying the mechanism by which benzene reaches the periodontal pocket—potentially acting as a reservoir would strengthen the manuscript. Furthermore, establishing whether a biochemical interaction between benzene and cotinine exists that enhances the inflammatory stimulus would add valuable insight.

Observations in form

Line 101 remove the script

Title of the table 2 must to be shorter

Comments on the Quality of English Language

I am not qualified to assess the quality of English.

Author Response

Thank you for your valuable feedback and insightful comments. Your input has helped us improve the clarity and quality of the manuscript.

We have carefully revised the manuscript in response to each reviewer's suggestions. All changes are highlighted in yellow in the revised manuscript and in blue in the responses below. We provide detailed replies to every comment for your review in a PDF document.

Sincerely,

Basel Hamoud

Reviewer 3 Report

Comments and Suggestions for Authors

The topic of the manuscript is interesting and fits the journal’s aims & scope.

The authors investigated the effect of blood benzene levels on periodontal disease severity. The study design is clearly described and corresponds to the set objectives. The main strength of the study is the large representative sample. Among the weaknesses the non-confirmed information about diabetes and other systemic diseases should be mentioned.

In general, the study is scientifically sound, and the conclusions are supported with the study results.

However, several issues should be corrected before publication.

  1. Double-check Table 1. There are the following mistakes/misprints:

Column 1. (1) Double-check Race Covariates.  (2) A misprint in the Age Covariate: it should be 50-69 instead of 45-69.

Column 2. It would be better to replace the column title: No Periodontitis instead of No Severe Periodontitis.

Column 4. There is a mistake in the number of patients with severe periodontitis. It seems it should be 796 instead of 769.

  1. Please, clarify the number of patients having increased blood benzene levels without increased blood cotinine levels and the number of patients increased blood benzene levels associated with increased blood cotinine levels
  2. Discussion section
  • Discossion partially repeats the Introduction in the same phrases. Please rephrase.
  • Based on the statistical analysis, the authors suggest that the effect of benzene on severe periodontitis is likely mediated through its impact on cotinine levels. However, in the Discussion there is no information related to the mechanisms of cotinine effect on periodontal disease. This information should be added to the Discussion too.

Author Response

(The authors gave the same response as above.)

Reviewer 4 Report

Comments and Suggestions for Authors

The study presents a noteworthy association between elevated blood benzene levels and severe periodontitis, highlighting an intriguing intersection between environmental exposure and oral health. However, to enhance the comprehensiveness and clarity of the findings, the following points should be addressed:

1, It is recommended that the authors categorize periodontitis into four distinct groups (no periodontitis, mild, moderate and severe) and present the classification in a tabulated format for clarity.

2, Diabetes mellitus is generally recognized as being significantly associated with severe periodontitis. However, your results did not show such a relationship. Please clarify this discrepancy.

3, Given that the general study population may not be familiar with benzene and its associated health risks, authors should:​

Provide Context: Explain the significance of benzene exposure and its known health effects, including its association with anemia and leukemia.​

Clarify Purpose: Detail why benzene blood tests are included in the study, especially if the purpose differs from traditional applications.​

Educate Readers: Offer background information on how benzene exposure is measured and the implications of the findings.​

By incorporating these explanations, authors can enhance the clarity and impact of their research, ensuring that readers fully comprehend the study's scope and significance.​

Author Response

(The authors gave the same response as above.)
